# 3D Thermal Network Supported by CF Felt for Improving the Thermal Performance of CF/C/Epoxy Composites

**DOI:** 10.3390/polym13060980

**Published:** 2021-03-23

**Authors:** Xinfeng Wu, Yuan Gao, Tao Jiang, Lingyu Zheng, Ying Wang, Bo Tang, Kai Sun, Yuantao Zhao, Wenge Li, Ke Yang, Jinhong Yu

**Affiliations:** 1College of Ocean Science and Engineering and Merchant Marine College, Shanghai Maritime University, Shanghai 201306, China; 201830410075@stu.shmtu.edu.cn (Y.G.); jiangtao9585@163.com (T.J.); 201740110013@stu.shmtu.edu.cn (L.Z.); 201940110009@stu.shmtu.edu.cn (Y.W.); luoyahangzhou@163.com (B.T.); kais@shmtu.edu.cn (K.S.); zhaoyt@shmtu.edu.cn (Y.Z.); wgli@shmtu.edu.cn (W.L.); 2School of Materials Science and Engineering, Central South University, Changsha 410083, China; 3Key Laboratory of Marine Materials and Related Technologies, Zhejiang Key Laboratory of Marine Materials and Protective Technologies, Ningbo Institute of Materials Technology & Engineering, Chinese Academy of Sciences, Ningbo 315201, China

**Keywords:** polymer composites, CF, thermal conductivity, 3D oriented structure

## Abstract

The heat generated by a high-power device will seriously affect the operating efficiency and service life of electronic devices, which greatly limits the development of the microelectronic industry. Carbon fiber (CF) materials with excellent thermal conductivity have been favored by scientific researchers. In this paper, CF/carbon felt (CF/C felt) was fabricated by CF and phenolic resin using the “airflow network method”, “needle-punching method” and “graphitization process method”. Then, the CF/C/Epoxy composites (CF/C/EP) were prepared by the CF/C felt and epoxy resin using the “liquid phase impregnation method” and “compression molding method”. The results show that the CF/C felt has a 3D network structure, which is very conducive to improving the thermal conductivity of the CF/C/EP composite. The thermal conductivity of the CF/C/EP composite reaches 3.39 W/mK with 31.2 wt% CF/C, which is about 17 times of that of pure epoxy.

## 1. Introduction

Over the past few years, the microelectronic industry has been rapidly developed and 5G technology is becoming more mature [1,2]. Electronic equipment is becoming more and more miniaturized and high-performance. Heat dissipation is still a serious factor hindering the development of electronic equipment. If the heat of electronic equipment cannot be discharged in time, the temperature of electronic equipment will rise rapidly, which will affect the efficiency and service life of electronic equipment. Therefore, the research and development of materials with high thermal conductivity has become a solution to the problem of heat dissipation. Polymer materials have good mechanical properties and chemical stability [3,4,5], but the thermal conductivity is generally low. Epoxy resin is an amorphous material, which mainly transmits heat through phonon heat conduction, so the thermal conductivity is only about 0.2 W/mK, so it needs to be modified [6].

The method of adding fillers is a fast way to improve the thermal conductivity of polymers. Metal materials (such as copper powder [7,8,9], silver powder [10,11], metal sheet and wire [12,13,14]), carbon materials (such as carbon fiber (CF) [15,16,17], graphene material [18,19], graphite material [20,21], carbon nanotubes [22,23], carbon black [24,25]), inorganic fillers (such as aluminum nitride [26,27], boron nitride [28,29,30], silicon nitride [31,32], silicon carbide [33], alumina [34,35]) and other high thermal conductivity fillers are often used to improve the thermal conductivity of polymers. When the particulate filler content is small, it is difficult to significantly improve the thermal conductivity of the matrix. When the filler forms a certain thermal network chain inside the polymer, the thermal conductivity of the composite material will be promoted significantly. However, when the content of the thermal conductive filler is too high, the granular material is prone to agglomeration, which makes the composite have certain defects and destroys the mechanical property. As a one-dimensional (1D) material, CF has high thermal conductivity in the axial direction of the fiber [36,37]. The CF filaments easily overlap each other inside the matrix to form a denser heat-conducting network [38,39]. The overlapping area between the CFs can greatly promote the heat conduction efficiency of the composite material. Therefore, the application and development of CF materials in the field of thermal management have become more and more prevalent in recent years. Tongle Xu et al. [40] mixed CF and graphene materials in a vacuum environment to form a three-dimensional thermal conductive structure. Then, the mixture was impregnated in polyamideimide resin (PAI). The thermal conductivity of CF composite material in the vertical direction is 0.53 W/mK, which is 1.65 times higher than that of pure PAI material. Jiake Ma et al. [41] fabricated epoxy/CF composites with a three-dimensional (3D) CF framework by the freeze-drying method. When the CF content is 13 vol%, the vertical thermal conductivity of the composite (2.84 W/mK) is about 15 times that of pure epoxy resin. Xiao Hou et al. [42] composited CF filaments with uniaxial orientation with Polydimethylsiloxane (PDMS), and the thermal conductivity of the material in the vertical direction reached 6.04 W/mK. The reed structure of CF promotes the transmission of heat, but the uniaxially oriented composite material also has the limits of the application in the field of thermal management materials.

Using CF to form a high thermal conductivity path network in composites is a research hotspot at present. In addition to forming an orientation structure in the composite, it is also necessary to form a connection between the orientation structures to further expand the heat path and promote the heat transfer efficiency of the heat conduction phonon. In this paper, CF felt with a three-dimensional CF framework structure was prepared by the “air flow network method” and the “needle-punching method”. The “graphitization process method” was used to fabricate CF/C felt with a carbon bonded structure between the three-dimensional CF frameworks by the pyrolysis of phenolic resin at high temperature. The CF orientation structure makes the CF have higher thermal conductivity in the orientation direction. The bonded carbon structure can further broaden the thermal conduction path, reduce the interfacial thermal resistance, and improve the thermal conductivity of the composites. Finally, the CF/C/Epoxy (CF/C/EP) composites were prepared by liquid-phase impregnation and the compression molding method. When the CF/C content reaches 35.2 wt%, the thermal conductivity of the composite reaches 3.39 W/mK. The methods used in this study are simple and can be used in large-scale production, which can provide some suggestions for the application of thermal conductive materials in the microelectronic and aerospace industries.

## 2. Materials and Methods

### 2.1. Materials

The SYT45 CF used in this study is a kind of Polyacrylonitrile-based CF (PAN CF). The diameter and the density of the CF material are 7 μm and 1.79 g/cm [3], respectively. The tensile strength and the tensile modulus of the CF are 4000 MPa and 230 GPa, respectively. The CF used in the study was purchased from Zhongfu Shenying CF Co., Ltd., Lianyungang, China. Phenolic resin PF-5408 with 65 wt% solid content, 9.5–11.8 wt% free phenol, 2.5–4.0 wt% moisture was supplied by Shengquan Group Co., Ltd., Jinan, China. Epoxy resin Araldite LY 1564 is a white transparent liquid with a viscosity of about 40 mPa·s. Polyamine curing agent Aradur 3486 is a light yellow liquid. Both reagents were supplied by Huntsman Company, Salt Lake City, Utah, USA.

### 2.2. Preparation Method of CF Felt

Figure 1 indicates the preparation process of the CF felt and CF/C felt. The CF felt is prepared by the airflow network technology and needle punching method. The main preparation steps are as follows: The CF filaments with a length of 12 mm were put into a chamber with some nets. After passing through a section of air duct, the CFs fell down to form an x-y structure CF mat. The gas pressure difference is one atmosphere. Then, the needle punching was introduced. Some of the CF filaments in the x-y plane were punched into the *z*-axis direction. The ratio of the CF filaments in the x-y plane and *z*-axis plane is 100:1. These *z*-axis CFs strengthened the connection of the mat. The density of the CF felt is about 0.04–0.08 g/cm [3]. The thickness is about 10 mm. This equipment is from Vulcan New Materials Technology Co., Ltd., Hangzhou, China.

### 2.3. Preparation Method of CF/C Felt

CF/C felt was optimized on the basis of CF felt. The main preparation methods are as follows: The phenolic resin and ethanol solution were mixed in a certain proportion and were stirred evenly. A quality of CF felt was immersed in a phenolic solution. The felt body was fully impregnated and then was transferred to the oven at 120 °C for 1 h to remove the ethanol, and 170 °C for 5 h to cure. At last, the cured composite was put into the high temperature furnace at 2000 °C for the graphitization process. The phenolic resin was decomposed under the high temperature and the carbon residue was left on the surface of the CF. The anisotropic CF/C composite has been prepared completely. This kind of composite is different from the CF felt. The CF/C felt, cured and graphitized, has a certain strength and 3D stable structure. The carbon produced by the pyrolysis of phenolic resin can bond the fibers together, increase the thermal conduction path between the fibers, and further improve the thermal conductivity of the composites.

#### Preparation Method of the CF/C/EP Composite

The preparation process of the CF/C/EP composite is shown in Figure 2. The physical picture is reflected in Figure 3. The raw materials of the CF/C/EP composite are the CF/C felt and epoxy resin. The main steps are as follows: Firstly, the CF/C felts were immersed into the epoxy system (epoxy/curing agent = 3/1) for 2 h in a vacuum oven. Secondly, the CF/C felt with epoxy resin was transferred into the hot press at 100 °C for 1 h under the pressure of 10 MPa. Then, the CF/C/EP composite was prepared after demolding. The size of the mold is 4 cm × 4 cm × 2 cm. In this paper, the content of CF/C felt in epoxy resin was increased by different compression ratios. The compressive ratios of the composite were 1, 1.5, 2 and 2.5, respectively.

### 2.4. Characterization Methods

In this study, digital analytical balance FA1106 and digital vernier caliper SHAHE were selected for measuring the mass and volume of the CF/C/EP composite. The two tools were manufactured by Biaogeda Laboratory Instrument Supplies Co., Ltd., Guangzhou, China and Ningbo Beilun District Kecheng Instrument Co., Ltd., Guangzhou, China, respectively. The optical photos of CF/C/EP composites were captured by a Sony α6000L digital camera. The microstructure of CF felt, CF/C felt and CF/C/EP composite materials was observed and analyzed by the scanning electron microscope. The instrument was produced by Jianhu Instrument Equipment Co., Ltd., Shanghai, China. The cross-section samples were obtained by the quenching method, and to make the image clear, a nano layer of gold material was evaporated on the cross section. The thermogravimetric analysis (TGA) of the CF/C/EP composite was conducted by the NETZSCH TG209 F3 from Germany under a nitrogen atmosphere at the range of 50–800 °C with a heating speed of 20 °C per min. The surface of the composite was tested by the thermal infrared imaging of Ti-400 from the Fluke Company, Everett, WA, USA. For studying the heat transfer performance of the composite, the thermal conductivity of samples was tested by the hot disk thermal constant analyzer (TPS500) according to ISO/DIS 22007-2.2 ”Plastics determination of thermal conductivity and thermal diffusivity—Part 2: Transient plane heat source (Hot Disk) method”, and this instrument was purchased from Kegonas Ltd., Stockholm, Sweden.

## 3. Results and Discussions

### 3.1. The Preparation and Morphology of the CF Composites

The schematic of the CF felt and CF/C felt is illustrated in Figure 4. The length of the CF used in the study is 12 mm. The 1D CF is distributed in the x-y plane by the “airflow network method”. Because of the non-binding between the CF filaments, the CF mat treated by the airflow is loose and difficult to be the felt. Moreover, the orientation of the few CF filaments could be changed with the introduction of the needle-punching method. The CFs in the x-y plane change the distribution of the CF to the *z*-axis direction, which enhances the bonding of the CF filaments in different x-y planes and provides more heat conduction paths.

In order to further increase the stability of CFs, pyrolysis carbon from phenolic resin was used to bond the CFs close to each other. The CF felt tends to be hard, which has a more sTable 3D structure. Due to the surface tension, the phenolic resin is more likely to stay at the overlapping point. The carbon residues mostly appear at the intersection of CF filaments, and the enhancement of the 3D structure provides a great help for the preparation of epoxy CF materials. It is worth noting that the carbon residue left by the phenolic resin strengthens the felt structure and further improves the conductivity of the composite. Oriented CFs can improve the in-plane thermal conductivity. The thermal conductivity of the 3D thermal skeleton can be improved by reducing the thermal resistance of the interface and widening the thermal conduction path.

As is shown in Figure 3, three samples are the pure epoxy resin, CF/C/EP composites with low content CF/C felt and high content CF/C felt. The cross-section of pure epoxy resin is smooth and transparent. However, the thermal conductivity of epoxy resin is low, and the introduction of CF can improve the thermal conductivity of the epoxy composites. It could be seen that when the CF/C content is low, some white holes exist in the cross-section of the CF/C composite, but these holes disappear when the CF/C content increases. The integrity of the composite is improved and more holes are squeezed out with the CF/C content increasing, which is good for the heat transmission inside the material.

Figure 5 reveals the morphology of the CF felt and the CF/C/EP composite. It can be seen in Figure 5 that the morphology of the composite in different directions is characterized by scanning electronic microscopy (SEM). The surface of the CF is smooth and the filament is about 7 μm at the diameter in Figure 5a. Many overlapping CF filaments are connected by the carbon residue in Figure 5b,c. The layered structure could remain after the immersing and heating process. As is shown in Figure 5b, there are many carbon residues formed after the treatment of the graphitization process. Therefore, these carbon residues further stabilize the structure of the CF/C felt and provide more heat conduction paths inside the composite. Figure 5c shows the SEM images of the CF/C felt in the *z*-axis direction, which provides a very clear layered structure. Most of the CFs are distributed in the x-y plane direction, and a small amount of CF filaments change the original distribution and present a vertical distribution with the introduction of the needle punching method. These CF filaments improve the stability to a certain extent, and play an important role in maintaining a good structure. On the other hand, the vertical CF is good for promoting the thermal conductivity of the material. Due to the effect of the surface tension, the phenolic resin with high viscosity easily stays at the intersection of CF filaments. After being immersed in the epoxy resin, the pores and cracks of the composite could almost not be found. Figure 5d–f shows the morphology images of the cross section of the CF/C/EP composite. The CF and epoxy resin are very tightly bonded together. The resin wraps the overlapping CFs, and there are almost no pores and defects between the two parts, which also reflects that the epoxy resin has been completely immersed during the impregnation process. As shown in Figure 5f, under the action of lamination, the contact area between CFs is greatly increased inside the composite material. Multiple CF filaments are tightly combined, which greatly reduces the interface thermal resistance. The possibility of contact increasing means that the heat conduction paths are enriched, and the phonons could spread smoothly in the carbon network, which optimizes the thermal performance of the composite. It is worth recognizing that the needle punching method and compression molding method are beneficial to improve the thermal performance of the CF/C/EP composite.

### 3.2. The Heat Resistance of the CF/C/EP Composite

Thermogravimetry can be used to characterize the thermal stability of a composite and the content of filler in the composite. Figure 6 is the curve of the TGA. Figure 6 exhibits the heat resistance and CF/C content in the CF/C/EP composite. Table 1 marks the final CF/C residue content in Figure 6. It can be found that the residues of CF/C/EP composites increase with the increasing CF/C felt content. When the compressive ratios of the CF/C felt are 1, 1.5, 2, and 2.5, the content of CF/C is 17%, 23.3%, 32.2% and 35.2%, respectively. It is worth noting that the CF/C/EP could not be decomposed until 280 °C, which shows the good heat resistance of the composite. In addition, with the increase in CF/C content in the composites, the decomposition temperature of the composites increases gradually, which indicates that the increase in CF/C is beneficial to improve the thermal decomposition temperature of the composites.

### 3.3. Thermal Conductivity of the CF/C/EP Composites

The thermal conductivity of the CF/C/EP composite is tested from two directions by a hot disk thermal constant analysis instrument. A thin disc-shaped double helix probe was used as a plane heat source and a temperature sensor, which was placed in the middle of the two testing samples with smooth surfaces. By adjusting the heating power and the temperature rise of the probe, the thermal conductivity of the composite could be measured. Figure 7a summarizes the thermal conductivity of the CF/C/EP composite. The thermal conductivities in the through-plane direction are 0.19 (0% CF/C), 0.71 (17.0% CF/C), 1.09 (23.3% CF/C), 1.22 (32.2% CF/C) and 1.34 W/mK (35.2% CF/C), respectively, showing that the thermal conductivity of the CF/C/EP composite increases with increasing CF/C content in the composite. Compared with the thermal conductivity in the through-plane, a more significant increase is shown in the in-plane direction. The thermal conductivities in the through-plane direction are 0.19 (0% CF/C), 2.00 (17.0% CF/C), 2.67 (23.3% CF/C), 3.28 (32.2% CF/C) and 3.39 W/mK (35.2% CF/C), respectively, showing that the thermal conductivity in the in-plane direction of the CF/C/EP composite also increases with increasing CF/C content in the composite. With the increase in CF content, the proportion of high thermal conductivity material (CF/C) increases, the contact surface between CFs increases, the thermal conduction path increases, and the thermal conductivity increases. Therefore, the thermal conductivities of CF/C/EP composites both in-plane and through-plane all increase with the increasing CF/C content. Although the contact interface between the CF/C skeleton and epoxy matrix also increases, there is a certain interface thermal resistance. However, the contribution of high thermal conductivity filler to the thermal conductivity is far higher than the negative effect of the interface, so the thermal conductivities of CF/C/EP composites all gradually increase with the increasing CF/C content.

Figure 7b reflects the enhancement of the thermal conductivity of the CF/C/EP composites with different CF contents. Comparing the two groups of data, it can be seen that the thermal conductivities of CF/C/EP composites in the in-plane direction are all much higher than those of CF/C/EP composites in the through-plane direction, which can be classified as the following points. (1) SEM images from Figure 5 show that most of the CFs are laid in the x-y plane, so most of the axial directions of CFs are distributed in the x-y plane. There are few CFs in the z direction. The thermal conductivity of CF in the axial direction is much higher than that of the vertical direction due to the effect of tensile crystal orientation, so it can be understood that the thermal conductivities of CF/C/EP composites in the in-plane direction are all much higher than those of CF/C/EP composites in the through-plane direction. (2) From the SEM images of Figure 5c, it can be seen that the distance of CFs between the adjacent planes is much higher than that in the same plane. Therefore, the connection points between the CFs in the adjacent planes are also relatively small, the heat conduction path is relatively small, and the CFs in the adjacent planes are blocked by a lot of epoxy resin, so the number of heat conduction systems in the z direction is relatively low. (3) It can be seen from Figure 5b,c that there are many bonding carbon points between the in-plane CFs, which can bond the in-plane CFs together and increase the thermal conductivity path, thus promoting the improvement of the thermal conductivity of CF/C/EP composites in the in-plane direction. However, due to the larger distance between the CFs in the adjacent planes, there are fewer bonding carbon points, and the increase in the thermal conduction path is relatively small, so the increase in thermal conductivity is relatively small. Bonded carbon points promote the in-plane thermal conductivity of the CF/C/EP composite. In conclusion, the three elements of CFs’ orientation structure, CFs’ spacing and bonding carbon points act on the composites together, which makes the enhancement of the thermal conductivity of CF/C/EP composites in the in-plane direction higher than that of CF/C/EP composites in the through-plane direction.

Figure 7c summarizes the data of the thermal conductivity from the previous research [41,42,43,44,45,46,47]. From Figure 7c, no matter whether the resin matrix of the composite is PDMS, epoxy or paraffin, the thermal conductivity of the composite gradually increases with the increasing CF content, which indicates that the high thermal conductivity filler CF can improve the thermal conductivity of the resin matrix. Through special methods such as the “freeze-drying orientation method” or other methods, CFs can be oriented so that the composites can obtain higher thermal conductivity in the orientation direction. Xiao hou [42] prepared PMDS/CF composites by the “freeze-drying method”. The short CFs in the composites were oriented in one direction. The thermal conductivity of the composites reached 6.04 W/mK (12.8 vol% CF). Jiake Ma [41] also prepared epoxy/CF composites by the “freeze-drying method”. The thermal conductivity of the composites reached 2.84 W/mK (13.0 vol% CF). However, the thermal conductivity of the composites is relatively high only in one direction, and the thermal conductivity in other directions is limited. In addition, the preparation methods of these composites are complex, so it is not suitable for the large-scale preparation of high thermal conductivity composites at this stage. In this paper, the preparation methods of high thermal conductivity composites have the following advantages. (1) 3D CF/C skeleton structure can make the composites obtain higher thermal conductivity in plane, which has certain performance advantages compared with other structures. The thermal conductivity of the CF/C/EP composite reaches 3.39 W/mK with 31.2 wt% CF/C, which is about 17 times that of pure epoxy. The x-y-z structure of the composite is consistent in all directions in the x-y plane, that is, higher thermal conductivity is obtained in both directions, which is conducive to heat dissipation in all directions in the x-y plane. (2) In this paper, CF/C felt was fabricated by CF and phenolic resin using the “airflow network method”, “needle-punching method” and “graphitization process method”. The CF/C/EP composites were prepared by the CF/C felt and epoxy resin using the “liquid phase impregnation method” and the “compression molding method”. These preparation methods are very mature and can be used to prepare high thermal conductivity composites on a large scale. The composites can be processed and applied directly. The maximum size of the composite can reach 2000 × 2000 × 50 mm, which is more conducive to industrialization.

### 3.4. Thermal Management Performance of the CF/C/EP Composite

The pure epoxy resin, low content and high content CF composites were selected as test subjects to study the thermal management performance. The surface temperatures of the composites were captured by thermal infrared imaging, as shown in Figure 8. The three samples all have the specification of 1 cm × 1 cm × 0.1 cm and were put on the same heat source. In Figure 8, the temperature distribution at the surface and the surface temperature curve with the heating time are shown.

The optical picture of the three samples, which are epoxy resin, low content CF/C and high content CF/C composite, respectively, is exhibited in Figure 8. It can be seen that the three samples reflect significantly different temperatures at the surface in 10 s. The temperature of the high content sample changes faster than that of the epoxy resin and the low content composite. As time goes on, the gap between the three samples becomes larger and larger. From the changes in the temperature, it can be found that the high content CF/C/EP composite has the best heat transfer and thermal management performance. The curve of the temperature trend at the surface can better reflect the thermal performance. The curve of temperature versus time can reflect the thermal management performance of the materials. The thermal conductivity of epoxy resin (0.19 W/mK) is the lowest, so the temperature of epoxy resin is the lowest among the three samples and the heat transfer ability of epoxy resin is the worst. With the addition of CF/C, the thermal conductivity of the composite increases gradually, and the ability of heat transfer of the composite material is gradually enhanced. Therefore, with the increasing CF/C content, the temperature of the composite increases more obviously. When heated for 40 s, the temperature at the surface of the epoxy resin, low content and high content composites is 95.4 °C, 113.3 °C and 141.2 °C, respectively.

Figure 9 displays the mechanism of the heat conduction in the CF/C/EP composite. CF/C forms a 3D thermal network in the epoxy resin. The higher thermal conductivity and better thermal management ability of the 3D thermal skeleton in the CF/C/EP composite can be attributed to the following points. (1) In the CF/C/EP composites, the heat transfer is along the axial direction of CF, so the thermal conductivity of CF/C/EP composites is relatively high, and the thermal management ability is good. The thermal conductivity of the CF in the axial direction is higher than that in the non-axial direction due to the tensile crystalline orientation of the CF in the manufacturing process. Seen from the SEM images in Figure 5, the axial direction of CFs is mainly concentrated in the x-y plane. When the heat is transferred in the plane, most of the heat transfer direction is along the axial direction of CFs. Therefore, the higher the CF/C content in the composite, the higher the thermal conductivity of the composite, the stronger the ability of heat dissipation, and the better the thermal management ability. (2) The bonding carbon points between the CFs increase the thermal conduction path, improve the thermal conductivity of the composite, and further improve the thermal management ability of the composite. According to the fractal theory [48,49,50], the increase in the connecting path is beneficial to improve the thermal conductivity of the composites. In this paper, the increase in carbon bonding is equivalent to the increase in the heat conduction path, so it is in line with the fractal theory. From the SEM images in Figure 5b, it can be found that CFs, especially the in-plane CFs, are bonded together by the bonded carbon produced by the high-temperature pyrolysis of phenolic resin, forming a heat conduction path. The ability of the composite to dissipate heat has been enhanced, and the ability of heat management has become better. (3) After compression, the density of the 3D skeleton structure is increased, the ability of heat transfer of the composite is enhanced, the thermal conductivity is increased, and the heat management ability is increased. After compression, the distance between CFs becomes smaller, the density of CFs and bonded carbon increases, and the heat conduction path increases obviously. Therefore, the thermal conductivity increases obviously, the heat transfer effect is more obvious, and the heat management ability is improved. From the above analysis, it can be seen that the three functions of the “CF axial increasing heat transfer effect”, “bonded carbon increasing heat conduction path effect” and “compression increasing heat conduction filler density effect” improve the thermal conductivity of the composite and the thermal management ability of the composite. It is hoped that the CF/C/EP composite structure with the 3D thermal conduction network can provide some suggestions for the application of CF composites.

## 4. Conclusions

The CF/C/EP composites with 3D thermal network structure were prepared by the CF/C felt and epoxy resin using the “liquid phase impregnation method” and the “compression molding method”. The CF/C felt has a 3D network structure, which is very conducive to improving the thermal conductivity of the CF/C/EP composite. The thermal conductivity of the CF/C/EP composite reaches 3.39 W/mK with 31.2 wt% CF/C, which is about 17 times that of pure epoxy. The “CF axial increasing heat transfer effect”, “bonded carbon increasing heat conduction path effect” and “compression increasing heat conduction filler density effect” can improve the thermal conductivity of the composite and the thermal management ability of the composite. The CF/C/EP composite structure with the 3D thermal conduction network can provide some suggestions for the application of CF composites in the field of aerospace and electronic material for thermal management.

## Figures and Tables

**Figure 1 polymers-13-00980-f001:**
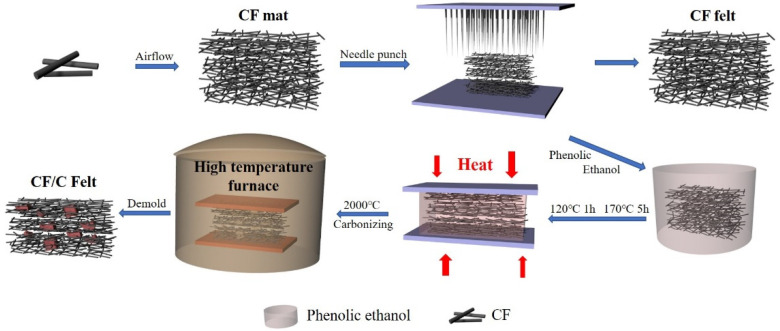
The preparation process of the carbon fiber (CF) felt and CF/C felt.

**Figure 2 polymers-13-00980-f002:**
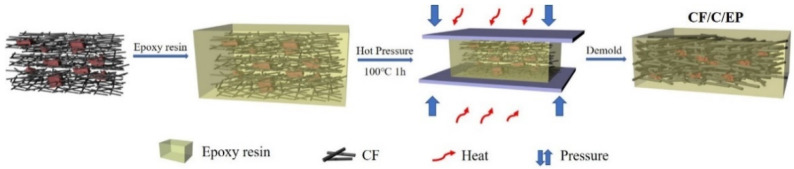
The preparation process of the CF/C/Epoxy (CF/C/EP) composite.

**Figure 3 polymers-13-00980-f003:**
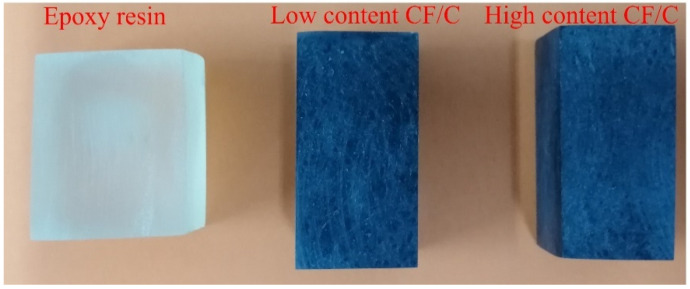
The physical image of the cured resin and the CF/C/EP composites.

**Figure 4 polymers-13-00980-f004:**
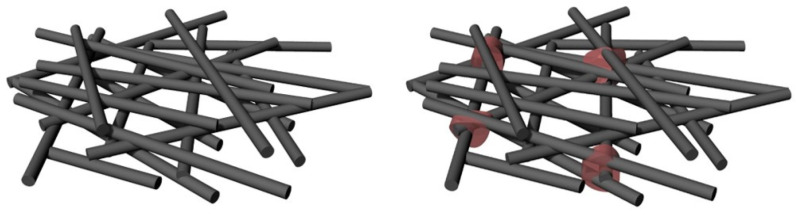
The schematic diagram of the CF felt and CF/C felt.

**Figure 5 polymers-13-00980-f005:**
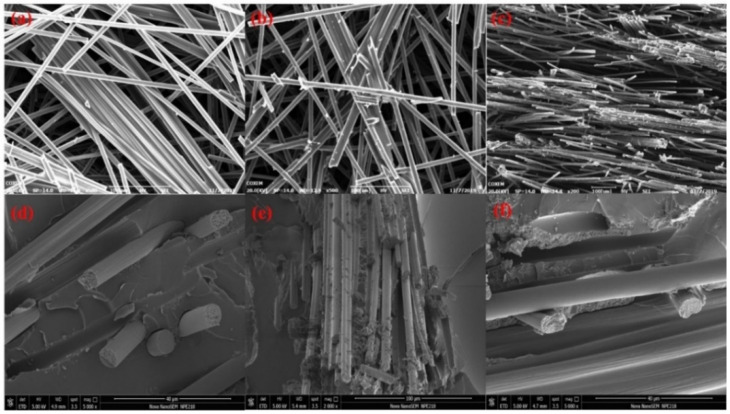
(**a**) The morphology of the CF felt in the in-plane direction, (**b**) the CF/C felt in the in-plane direction, (**c**) in the through-plane direction, (**d**–**f**) the CF/C/EP composite.

**Figure 6 polymers-13-00980-f006:**
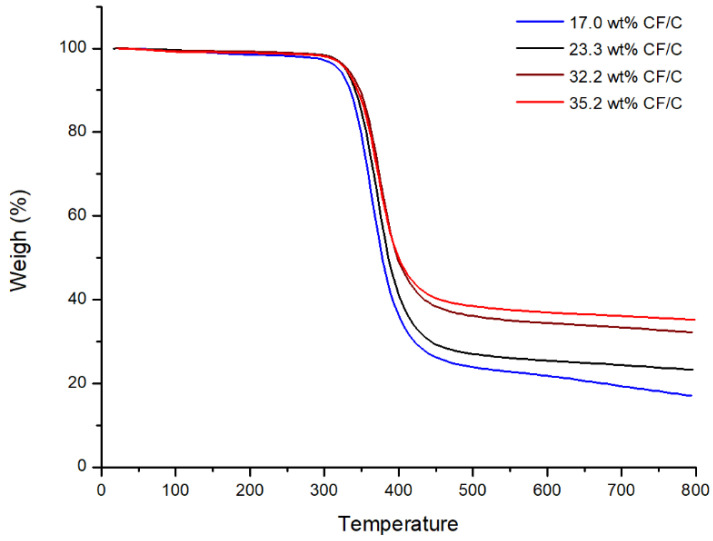
The curve of the TGA.

**Figure 7 polymers-13-00980-f007:**
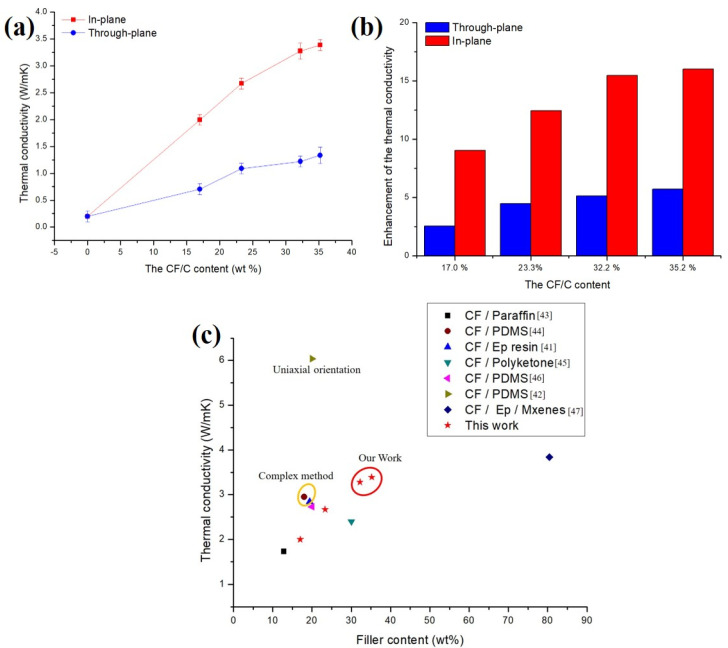
(**a**) The thermal conductivity of the CF/C/EP composite, (**b**) the enhancement of the thermal conductivity of the CF/C/EP composites with different CF/C contents, (**c**) the data of the thermal conductivity from previous research.

**Figure 8 polymers-13-00980-f008:**
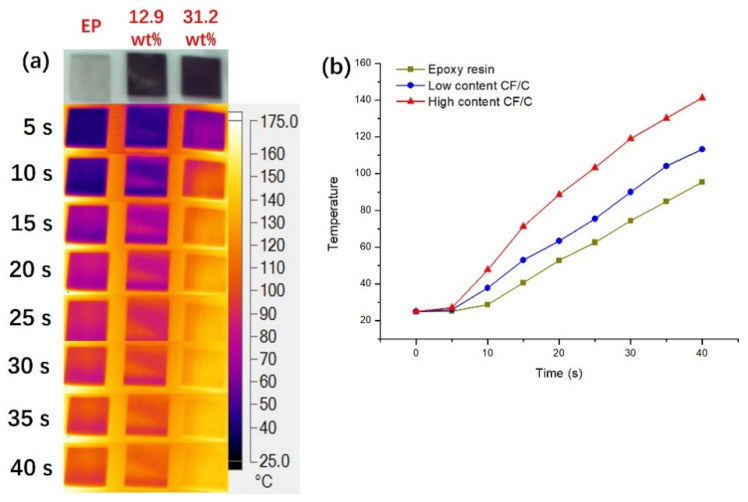
(**a**) Infrared thermal image, (**b**) the surface temperature of different CF/C/EP composites with time during heating.

**Figure 9 polymers-13-00980-f009:**
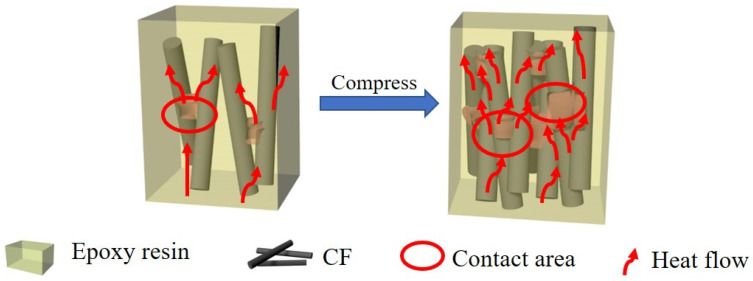
The mechanism of the heat conduction in the CF/C/EP composite.

**Table 1 polymers-13-00980-t001:** The loading of the CF/C in the various CF composites.

Samples	CF/C [vol%]	CF/C [wt%]
CF/C/EP-1	11.3	17.0
CF/C/EP-1.5	15.8	23.3
CF/C/EP-2	22.8	32.2
CF/C/EP-2.5	25.3	35.2

## Data Availability

The data presented in this study are available on request from the corresponding author.

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
