# Peer review of "3D Thermal Network Supported by CF Felt for Improving the Thermal Performance of CF/C/Epoxy Composites"

_polymers, 2021, doi:10.3390/polym13060980_

Round 1
Reviewer 1 Report
In Ref. 49, “Fractals-Complex Geometry Patterns and Scaling in Nature and Society 2020, 28 (5)” should be corrected as “Fractals 2020, 28, 2050080”; In Ref. 50, “Fractals-Complex Geometry Patterns and Scaling in Nature and Society 2020, 28 (2)” should be corrected as “Fractals 2020, 28, 2050029”;
Author Response
Point 1: In Ref. 49, “Fractals-Complex Geometry Patterns and Scaling in Nature and Society 2020, 28 (5)” should be corrected as “Fractals 2020, 28, 2050080”; In Ref. 50, “Fractals-Complex Geometry Patterns and Scaling in Nature and Society 2020, 28 (2)” should be corrected as “Fractals 2020, 28, 2050029”;
Response: Thank you for your advice. This mistake has been corrected in the paper.
- Xiao, B.; Zhang, Y.; Wang, Y.; Wang, W.; Chen, H.; Chen, X.; Long, G., AN INVESTIGATION ON EFFECTIVE THERMAL CONDUCTIVITY OF UNSATURATED FRACTAL POROUS MEDIA WITH ROUGHENED SURFACES. Fractals-Complex Geometry Patterns and Scaling in Nature and Society 2020, 28, 2050080.
- Xiao, B.; Wang, S.; Wang, Y.; Jiang, G.; Zhang, Y.; Chen, H.; Liang, M.; Long, G.; Chen, X., EFFECTIVE THERMAL CONDUCTIVITY OF POROUS MEDIA WITH ROUGHENED SURFACES BY FRACTAL-MONTE CARLO SIMULATIONS. Fractals-Complex Geometry Patterns and Scaling in Nature and Society 2020, 28, 2050029.

Reviewer 2 Report
Now authors responce my comment well. Additionally, eqipments used to realize airflow network technology and needle punching method should be described in details, including equipment schemes and all the opeating parameters.
Author Response
Point 1: Now authors responce my comment well. Additionally, eqipments used to realize airflow network technology and needle punching method should be described in details, including equipment schemes and all the opeating parameters.
Response: Figure 1 indicates the preparation process of the CF felt and CF/C felt. The CF felt is prepared by the airflow network technology and needle punching method. The main preparation steps are as follows: The CF filaments with a length of 12 mm were put into a chamber with some nets. After passing through a section of air duct, the CFs fell down to form an x-y structure CF mat. The gas pressure difference is one atmosphere. Then the needle punching was introduced. Some of the CF filaments in the x-y plane were punched into the z-axis direction. The ratio of the CF filaments in the x-y plane and z-axis plane is 100:1. These z-axis CF strengthed the connection of the mat. The density of the CF felt is about 0.04-0.08 g/cm3. The thickness is about 10 mm. This equipment is from Hangzhou Vulcan New Materials Technology Co., Ltd.

This manuscript is a resubmission of an earlier submission. The following is a list of the peer review reports and author responses from that submission.
Round 1
Reviewer 1 Report
Paper presents any idea to create the thermal conductive net in epoxy based composites. Before further consideration paper should be revised in accordance with the following comments:
- English should be improve throw the manuscript, as pater contains a lot of typo and grammar errors, such as “compositehas” (section 2.3.1), “ploymer” (p. 5), “dierction“ (Fig. 5, caption) and so on.
- The sentence “Thermal conductive fillers mainly include the inorganic non-metallic materials, usually oxides, nitrides and carbides” generally is not true, because of metallic fillers, such as copper or aluminum, possess higher thermal conductivity that ceramics, and, correspondingly, are more attractive to utilization as fillers to improve polymer composite thermal conductivity. Please revise this sentence.
- The Idea of C felt should be described and discussed in Introduction.
- Section 2.1. Raw CF should be described in more details.
- Section 2.2. Airflow network technology should be described in detail.
- Section 2.3. What are the small brown cylinders in the last picture of Figure 1? Please describe it in the text in detail.
- Section 2.3.1. What means “…felt with different content…”? Please describe the procedure more clear.
- Section 2.4. Please describe the thermal conductivity measurement procedure in detail. How the thermal capacity Cp was measured in your experiments?
Author Response
Response to Reviewer 1
Point 1: English should be improve throw the manuscript, as pater contains a lot of typo and grammar errors, such as “compositehas” (section 2.3.1), “ploymer” (p. 5), “dierction“ (Fig. 5, caption) and so on.
Response: Thank you for your advice. The mistakes have been corrected in the paper.
Point 2: The sentence “Thermal conductive fillers mainly include the inorganic non-metallic materials, usually oxides, nitrides and carbides” generally is not true, because of metallic fillers, such as copper or aluminum, possess higher thermal conductivity that ceramics, and, correspondingly, are more attractive to utilization as fillers to improve polymer composite thermal conductivity. Please revise this sentence
Response: Thank you for your advice. The mistakes have been corrected in the paper.
Point 3: The Idea of C felt should be described and discussed in Introduction
Response: Thank you for your advice. The Idea of C felt has been described and discussed in Introduction. In this study, CF felt which was only composed of the CF fialments was prepared by the airflow network and needle-punching method.
Point 4: Section 2.1. Raw CF should be described in more details.
Response: Thank you for your advice. Raw CF has been described in more details in the paper. The length of the PAN-based CF used in this study is about 12 mm. The density of the CF is 1.8 g/cm3 and the specification is SYT45, it was purchased from Zhongfu Shenying CF Co., Ltd, China.
Point 5: Section 2.2. Airflow network technology should be described in detail.
Response: Thank you for your advice. The Airflow network technology has been described in detail in the paper. The CF filaments with a length of 12 mm were put into a chamber with some nets. The air flow was introduced with the high-pressure evacuation equipment. The CF could move along with the flow. Then the evacuation equipment was turned off and the CF landed in the chamber.
Point 6: Section 2.3. What are the small brown cylinders in the last picture of Figure 1? Please describe it in the text in detail.
Response: Thank you for your advice. The small brown cylinders in the last picture of Figure 1 has been described in the figure.
Point 7: Section 2.3.1. What means “…felt with different content…”? Please describe the procedure more clear.
Response: Thank you for your advice.The explanation has been added in the paper.
Point 8:Section 2.4. Please describe the thermal conductivity measurement procedure in detail. How the thermal capacity Cp was measured in your experiments?
Response: Thank you for your advice. The details has been added in the section 3.3. A thin disc-shaped double helix probe was used as a plane heat source and a temperature sensor, which was placed in the middle of the two test sample with the smooth surface. The temperature rised after the probe is energized, the thermal resistance coefficient of the probe changed with the temperature change, the temperature rise and response time were measured, and the thermal conductivity was obtained.
Reviewer 2 Report
The authors of the study have centered on the fabrication of CF/carbon felt (CF/C felt) by CF and phenolic resin using “airflow network method” - “needle-punching method” - “graphitization process method”. The CF/C/Epoxy composites (CF/C/Ep) were prepared by the CF/C felt and epoxy resin using “liquid phase impregnation method” and “compression molding method”. The authors have then shown that the CF/C composite has a 3D network structure, which is very conducive to improving the thermal conductivity. The thermal conductivity of the CF/C/Ep composite reached 3.39 W/mK with 31.2 wt% CF/C, which is about 17 times that of pure epoxy.
In fact, the research reported is recurrent. The conclusions drawn are in line with the texts discussed in the main body of the ms. Pictures provided are of publication quality. The work deserves publication in polymers. However, I suggest the authors to consider the following and revise their paper.
1- Remove first 6 lines from abstract (from top) and make it more compact.
2-The last paragraph of Introduction must be rewritten, with clear objectives. "Remove "In this experiment" - rather say "In this study,.."
3- The section "Results" deals with results and discussion both. So, the subtitle "Results" should be revised. Otherwise, authors should add another section called "Discussion".
Author Response
Response to Reviewer 2
Point 1: Remove first 6 lines from abstract (from top) and make it more compact.
Response: Thank you for your advice.The abstract has been revised.
Point 2: The last paragraph of Introduction must be rewritten, with clear objectives. "Remove "In this experiment" - rather say "In this study,.."
Response: Thank you for your advice.The last paragraph of Introduction has been rewritten.
Point 3: The section "Results" deals with results and discussion both. So, the subtitle "Results" should be revised. Otherwise, authors should add another section called "Discussion".
Response: Thank you for your advice. The "Discussion" has been added.
Reviewer 3 Report
The heat generated by high-power device will seriously affect the operating efficiency and service life of the devices, which greatly limits the development of the microelectronic field. Heat transfer is important in wide fields of engineering and sciences. Heat conduction is the transfer of energy between near molecules in an object as a result of temperature gradient. Heat transfer in carbon fiber/porous materials is the reflection of the details of porous structure. Furthermore, heat transfer is estimated by the physical mechanisms of the basic transport procedures within the distrinct phases of carbon fiber/ porous materials and thermal exchange in a interface. The prediction of the thermal conductivity of carbon fiber/porous materials has drawn the attention of many researchers, because it is a critical transport parameter characterized by the geometric structure of carbon fiber/porous materials. In this manuscript, CF/carbon felt (CF/C felt) were fabricated by CF and phenolic resin using “airflow network method” - “needle-punching method” - “graphitization process method”. Then the CF/C/Epoxy composites (CF/C/Ep) were prepared by the CF/C felt and epoxy resin using “liquid phase impregnation method” and “compression molding method”. The topic is important, the results are interesting and the methodology followed is appropriate, while the content falls well within the scope of this Journal. In general the paper makes fair impression and my recommendation is that it merits publication in this Journal, after the following major revision:
- The authors need to reorganize the current introduction, which normally consists of three parts at least: background, literature review, brief of the proposed work. The current one is nothing but a literature review. Why their work is important comparing to previous reports? I think this is essential to keep the interest of the reader.
- In Fig.6, 7(a), 7(c) and 9(b), the authors should give the explanations for the difference of data collected from different sources.
- Experiment part. Although the results look “making sense”, the current form reads like a simple lab report. The authors should dig deeper in the results by presenting some in-depth discussion.
- The research results showed that the CF/C composite has a 3D network structure, which is very conducive to improving the thermal conductivity. The thermal conductivity of the CF/C/Ep composite reached 3.39 W/mK with 31.2 wt% CF/C, which is about 17 times that of pure epoxy. The authors should give some explanation on above results and data and analyze the physical mechanism in detail.
- With the development of the electronics industry (see [International Journal of Heat and Mass Transfer, 2019, 137:365-371]), more devices tend to be miniaturized and integrated. The proposed work mainly focuses on lab work. It does not necessarily imply that the theoretic work (modeling) is not important. The authors omit this part during the current literature review, which should include a brief review of the theoretic work after revision. In the theoretic perspective, fractal theory is a very important tool, which can investigate the thermal conductivity of carbon fiber/porous materials (see [Fractals, 2020, 28(2): 2050029; Fractals, 2020, 28(5): 2050080]). Authors should introduce some related knowledge to readers.
- English grammar and syntax has to be checked carefully throughout the manuscript. There are several grammatical mistakes in the manuscript and it is very difficult to follow anything if they are not corrected.
Author Response
Response to Reviewer 3
Point 1: The authors need to reorganize the current introduction, which normally consists of three parts at least: background, literature review, brief of the proposed work. The current one is nothing but a literature review. Why their work is important comparing to previous reports? I think this is essential to keep the interest of the reader.
Response: Thank you for your advice.The abstract has been revised in the paper.
Point 2: In Fig.6, 7(a), 7(c) and 9(b), the authors should give the explanations for the difference of data collected from different sources.
Response: Thank you for your advice. More explanation has been added in the paper. The carbon residue in the CF/C/EP composite were added with the CF felt content increasing. It could be seen that the mass of all the composite decreased slightly when the temperature reached 60℃.
Moreover, compared with the blue line, a more significant increase was found in the thermal conductivity of the composite in the in-plane direction. From the figure 7(a), the red line of the thermal conductivity could reflect the sharp increase and when the CF content are 12.9%, 19.54%, 24.92% and 31.15%, the thermal conductivity in the in-plane direction are 2.00, 2.67, 3.28 and 3.39 W/mK respectively.
By analyzing the previous datas, athough the polymers used to composite with the CF are different, the thermal conductivity also has the relationship with the filler content. The rough rule of the figure 7(c) indicated that the thermal conductivity increased with the fillers adding.
It is worth noted in figure 9b that that the epoxy resin showed the slowest trend at the surface temperature and the lowest efficiency of the heat transfer.
Point 3: Experiment part. Although the results look “making sense”, the current form reads like a simple lab report. The authors should dig deeper in the results by presenting some in-depth discussion.
Response: Thank you for your advice. Some contents have been added in the experiment part.
Point 4: The research results showed that the CF/C composite has a 3D network structure, which is very conducive to improving the thermal conductivity. The thermal conductivity of the CF/C/Ep composite reached 3.39 W/mK with 31.2 wt% CF/C, which is about 17 times that of pure epoxy. The authors should give some explanation on above results and data and analyze the physical mechanism in detail.
Response: Thank you for your advice. Some explanation has been written in the section 3.3.
Point 5: With the development of the electronics industry (see [International Journal of Heat and Mass Transfer, 2019, 137:365-371]), more devices tend to be miniaturized and integrated. The proposed work mainly focuses on lab work. It does not necessarily imply that the theoretic work (modeling) is not important. The authors omit this part during the current literature review, which should include a brief review of the theoretic work after revision. In the theoretic perspective, fractal theory is a very important tool, which can investigate the thermal conductivity of carbon fiber/porous materials (see [Fractals, 2020, 28(2): 2050029; Fractals, 2020, 28(5): 2050080]). Authors should introduce some related knowledge to readers.
Response: Thank you for your advice.The heat conduction mechanism of the carbon residue has been explained in the paper. The mechanism of the heat conduction in the CF/C/EP composite with compressive process has been written in paper.
Point 6: English grammar and syntax has to be checked carefully throughout the manuscript. There are several grammatical mistakes in the manuscript and it is very difficult to follow anything if they are not corrected.
Response: Thank you for your advice. Some mistakes have been corrected in the paper.

Round 2
Reviewer 1 Report
Authors answer my comments partially, most of comments were not answered satisfactorily, namely:
Points 1, 3, 5 were nearly answered well.
Point 2 was answered not correct because of following disadvantages:
- Indeed, not all the oxides, nitrides and carbides possess high thermal conductivity, so this sentence is wrong.
- Thermal conductivity of oxides, nitrides and carbides is higher than what? It is not clear what you means here.
- It is not clear, why, in your opinion, non-metallic material are more attractive fillers than metallic one. Please describe it!
Point 4. This answer is not enough. Please give basic properties (mechanical, physical) of raw CF, also the CF diameter should be given. SEM micrographs of raw CF are required also.
Point 6. I find no required description in figures. In any cases, detailed description should ne given in section 3.2 text also.
Point 7 was not answered. What is “CF/C felt with different CF content”, when it appear? No data on the CF content in CF/C felt was provided in section 2.3.
Point 8. Details of experiment, including trademark of equipment and standard (ISO) used for thermal conductivity measurements should be given in section 2.4.
Reviewer 3 Report
The revised form simply ignore my comments made in the first round, thus I cannot commend its publication.